# Long-Term Associations between Human Cytomegalovirus Antibody Levels with All-Cause Mortality and Cardiovascular Outcomes in an Australian Community-Based Cohort

**DOI:** 10.3390/v14122676

**Published:** 2022-11-29

**Authors:** Silvia Lee, Nikki van den Berg, Alison Castley, Mark Divitini, Matthew Knuiman, Patricia Price, David Nolan, Frank Sanfilippo, Girish Dwivedi

**Affiliations:** 1Harry Perkins Institute of Medical Research, University of Western Australia, Murdoch 6150, Australia; 2Pathwest Laboratory Medicine, Department of Microbiology, Perth 6009, Australia; 3Department of Laboratory Medicine, Radboud University, 6525 GA Nijmegen, The Netherlands; 4Department of Clinical Immunology, Royal Perth Hospital, Perth 6000, Australia; 5School of Population and Global Health, University of Western Australia, Perth 6009, Australia; 6Curtin Medical School, Curtin University, Bentley 6102, Australia; 7Department of Cardiology, Fiona Stanley Hospital, Murdoch 6150, Australia

**Keywords:** human cytomegalovirus, cardiovascular disease, Busselton Health Survey, outcomes

## Abstract

Human cytomegalovirus (HCMV) infection has been shown to increase the risk of cardiovascular events and all-cause death among individuals with clinically apparent cardiovascular disease (CVD). Whether this association exists in individuals with no history of CVD remains unclear. Serum levels of HCMV IgG antibody were measured using an ELISA in 2050 participants aged 40–80 years from the 1994/1995 Busselton Health Survey who did not have CVD at baseline. Outcomes were all-cause death, cardiovascular death, acute coronary syndrome (ACS) and major adverse coronary and cerebrovascular events (MACCE, composite of all-cause death, ACS, stroke and coronary artery revascularisation procedures). Cox proportional hazards regression analysis was used to investigate HCMV antibody levels as a predictor of death and cardiovascular outcomes during follow-up periods of 5, 10 and 20 years. At baseline, participants had a mean age of 56 years and 57% were female. During the 20-year follow-up, there were 448 (21.9%) deaths (including 152 from CVD), 139 (6.8%) participants had ACS and 575 (28.0%) had MACCE. In the fully adjusted model, levels of HCMV antibody at 20 years was associated with all-cause death (HR 1.04; 95% CI 1.00, 1.07, *p* = 0.037) but not with CVD death, ACS or MACCE. Levels of HCMV antibody are associated with all-cause death but not with cardiovascular outcomes in adults without pre-existing CVD.

## 1. Introduction

Cardiovascular disease (CVD) is one of the leading health problems worldwide, contributing to approximately 17.9 million deaths in 2015 [1]. Atherosclerosis is a common cause of CVD and is characterised by the accumulation of lipids in the arterial walls triggering chronic inflammation [2]. The subsequent narrowing of the arteries can lead to coronary artery disease and stroke. Several pathogens can trigger atherosclerosis, including human cytomegalovirus (HCMV) and Chlamydia pneumoniae [3]. HCMV is a beta-herpesvirus that generally establishes asymptomatic persistent infections in immunocompetent adults, but intermittent reactivations and re-infections can occur throughout life and may be symptomatic [4]. HCMV can generate diverse clinical syndromes in those with a weakened immune system (e.g., transplant recipients and people living with HIV) [5]. The estimated global seroprevalence is 83% in the general population [6]. In Australia, the HCMV seroprevalence is approximately 70% in individuals aged over 40 years [7].

Previous epidemiological studies have yielded conflicting results on the associations between HCMV infection and cardiovascular outcomes. A meta-analysis of 30 studies (3328 patients with atherosclerosis and 2090 controls) demonstrated an increased risk of atherosclerosis in people with HCMV infection [8]. Similarly, a meta-analysis of 10 community-based prospective studies (34,564 participants and 4789 patients with CVD) reported an association between HCMV seropositivity and risk of future CVD events [9]. Furthermore, in a longitudinal study of individuals followed for 27 years, the prevalence of CVD was associated with HCMV seropositivity and elevated levels of HCMV antibody [10]. However, a recent study of over 10,000 individuals from five cohorts of community-dwelling older adults found no association between HCMV serostatus or antibody levels with cardiovascular mortality [11].

Overall, most studies were performed in populations with clinically apparent CVD, and individuals were either stratified according to HCMV serostatus or antibody levels dichotomised into tertiles. Therefore, we investigated whether levels of HCMV antibody examined as a continuous variable better revealed associations with all-cause mortality and cardiovascular outcomes over a 20-year period in adults with no pre-existing CVD. This was achieved using a well-characterised community-based cohort recruited in Busselton, Western Australia.

## 2. Materials and Methods

### 2.1. Study Participants and Data

The study included a retrospective cohort of 2050 individuals aged 40–80 years who participated in the 1994/95 survey of the Busselton Health Study and had no history of cardiovascular disease. These data were linked to matching records of hospital admissions from all public and private hospitals in Western Australia, and death registry data, obtained from the Western Australian Department of Health. A history of CVD was identified from the linked data of hospital admissions during the 15 years before the survey. All participants provided written informed consent at the time of the 1994/95 survey and the study was approved by the Human Ethics Research Committees of the Western Australian Department of Health (2011/60) and the University of Western Australia (RA/4/1/6682).

### 2.2. Clinical Variables

Variables available from the clinical dataset and considered as confounders in regression models included age, gender, smoking status (never, ex or current), body mass index (BMI), C-reactive protein (CRP), comorbidities (diabetes, chronic obstructive pulmonary disease (COPD)), lipids (total cholesterol, high-density lipoprotein (HDL) cholesterol and triglycerides), estimated glomerular filtration rate (eGFR), blood glucose, systolic blood pressure (SBP) and antihypertensive medications. Clinical assessments were completed in the 1994/95 survey. Blood pressure was measured using a mercury sphygmomanometer after 5 min rest in a sitting position. Body mass index was defined as weight (kg) divided by height (m) squared. Serum lipids, glucose, CRP and creatinine were determined from a fasting blood sample at the time of survey. Details of smoking, alcohol consumption, diabetes, COPD and use of antihypertensive drugs were obtained by questionnaire.

### 2.3. Measurement of HCMV-Reactive IgG Antibodies, sCD14 and sCD163

Fasting blood samples were obtained from participants during the 1994/95 survey, and stored at −70 °C after processing. Aliquots were obtained for use in the current study for those who had a sufficient quantity of stored sample remaining. Antibody levels to HCMV were determined using an established in-house enzyme-linked immunosorbent assay (ELISA) as previously described [12]. Briefly, HCMV-reactive IgG antibodies were detected using lysate of human foreskin fibroblasts (HFF) infected with the laboratory strain AD169. Ninety-six-well microtitre plates were coated with lysate overnight at 4 °C. Plates were washed with 0.05% Tween/phosphate buffered saline (PBS) and blocked with 5% bovine serum albumin (BSA)/PBS. Serum samples pre-diluted in 2% BSA/PBS were serially diluted alongside a standard plasma from a HCMV-seropositive healthy donor assigned a value of 1000 arbitrary units (AU/mL). After 2 h incubation at room temperature, horseradish peroxidase conjugated anti-human IgG (Sigma–Aldrich, New South Wales, Australia) diluted 1:4000 in 2% BSA/PBS was added for 1 hour at room temperature. Tetramethylbenzidine substrate (Sigma–Aldrich) was added and colour development was stopped with 1 M H_2_SO_4_. Plates were read at 450 nm using an Omega plate reader (BMG Labtech, Victoria, Australia). The standard curves from HCMV lysate-coated plates were used to quantify antibodies bound to HFF lysate-coated plates. The calculated values were then subtracted from data obtained with HCMV. Samples achieving HCMV antibody levels greater than three standard deviations above the mean (3475 AU/mL) of nine seronegative samples, assessed by the ARCHITECT CMV IgG assay (Abbott Diagnostic Systems, Lake Bluff, IL, USA) and included in the in-house ELISA, were defined as seropositive. The inter-assay coefficient of variance for the assay was 9.4%. Levels of soluble biomarkers, sCD14 and sCD163, were quantitated using Quantikine ELISA kits (R&D Systems, MN, USA) according to the manufacturer’s instructions [13].

### 2.4. Outcomes

Outcome events for the period from baseline survey to 2014 were determined using linked hospital admissions and death data. Primary endpoints included all-cause death, cardiovascular death, acute coronary syndrome (ACS) and major adverse coronary and cerebrovascular events (MACCE, composite of all-cause death, ACS, stroke and coronary artery revascularisation procedures). A secondary analysis was performed, which only included cardiovascular deaths (instead of all-cause death) in the MACCE definition. International Classification of Diseases 9th revision clinical modification (ICD-9-CM) codes were used for events up to 30 June 1999 and the 10th revision Australian Modification (ICD-10-AM) was used for events from 1 July 1999 onwards. Cardiovascular events were defined as hospital admissions with a diagnosis of coronary heart disease (ICD-9-CM 410-414; ICD-10-AM I20-25) or stroke (ICD-9-CM 430-437; ICD-10-AM I60-68) or death from CVD (ICD-9-CM 390-459; ICD-10-AM I00-99).

### 2.5. Statistical Analysis

Descriptive statistics are presented as mean (standard deviation) for quantitative variables and percentage for categorical variables. Associations between serum HCMV antibody levels with death and cardiovascular outcomes were determined using Cox proportional hazards regression modelling. To investigate whether associations varied over time, separate proportional hazards models were assessed at 5-, 10-, and 20-year follow-ups. HCMV was examined as a continuous variable and due to the skewed distribution of serum levels, we used their natural log transform in the regression models. The estimated hazard ratios (HRs) with their 95% confidence interval (CI) were reported after adjustment for baseline confounders. Three levels of adjustment were used. Model 1 adjusted for age and sex. Model 2 further adjusted for smoking, BMI, BP treatment, SBP, diabetes, total cholesterol, HDL cholesterol, triglycerides, blood glucose and CRP. Model 3 further adjusted for COPD and eGFR. As monocytes are the primary targets for HCMV and a site of viral latency and persistence [4], model 4 included sCD14 and sCD163 as markers of their activation. HCMV antibody levels or HCMV viremia (indicative of reactivation) have been shown to correlate with levels of these biomarkers [14,15,16]. Statistical analyses were performed using SAS 9.4 (SAS Institute Inc., Cary, NC, USA). *p*-values < 0.05 were considered statistically significant.

## 3. Results

Baseline characteristics of the Busselton study participants are summarised in Table 1. The study cohort consisted of 2050 individuals, with a mean age of 56 years and 57% were female. The mean BMI was 26.6 kg/m^2^, 10.7% were smokers, 6.4% had diabetes and 16.8% of individuals were taking antihypertensive medication. Overall, the mean log HCMV antibody level was 8.59 (SD 4.12) AU/mL. Prevalence of HCMV seropositivity in the cohort was 72%.

During the 20-year follow-up period, there were 448 (21.9%) deaths including 152 (7.4%) deaths attributable to CVD. The top 10 leading causes of death are outlined in Appendix A. A total of 139 (6.8%) participants had ACS, and 575 (28.0%) people had MACCE (with cardiovascular death), whilst 319 (15.6%) had MACCE (with cardiovascular death).

Levels of HCMV antibody were associated with all-cause death at 20 years (Model 1: HR, 1.04; 95% CI 1.01–1.08; *p* = 0.013, Table 2), but not at 5- and 10-year follow-ups (Appendix A) after adjustment for age and sex. The association remained unchanged after further adjustment for smoking, BMI, BP treatment, SBP, diabetes, total cholesterol, HDL cholesterol, triglycerides, blood glucose and CRP (Model 2). Likewise, there were no significant associations after adjustments for COPD and eGFR (Model 3) and for sCD14 and sCD163 (Model 4). The failure of adjustments for sCD14 and sCD164 to modify associations between all-cause mortality and HCMV antibodies may reflect the weak correlations seen between HCMV antibody levels and sCD14 (r = 0.05, *p* = 0.01) and sCD163 (r = 0.07, *p* = 0.0004) levels. HCMV antibody levels were not associated with cardiovascular death, ACS and MACCE (with cardiovascular death or all-cause death) (Table 2).

## 4. Discussion

HCMV infection has been implicated in the pathogenesis of CVD and other chronic diseases associated with ageing [17]. The present study represents the longest follow-up of any study evaluating HCMV antibody levels and shows that HCMV antibody levels were a predictor of all-cause death in middle-aged and older adults from the general population over a 20-year period. This is consistent with previous shorter follow-up studies. In a US population of 14,153 individuals aged 25–90 years followed for a mean duration of 13 years, HCMV seropositivity was associated with a 19% higher risk of all-cause death [18]. Similarly, in a group of 511 older people aged 65–94 years, HCMV seropositivity was associated with a 35% increase in mortality during an 18 year follow-up period [19]. In contrast, Chen et al., found no association between HCMV seropositivity and all-cause death in 10,122 community-dwelling older adults aged between 59–93 years [11]. However, the authors did observe increased mortality in individuals in the highest HCMV IgG quartile compared to seronegative subjects, but this was no longer evident after adjustment for BMI, education, smoking status, number of comorbidities, medications and C-reactive protein. In our study, the association between HCMV antibody levels and all-cause mortality remained significant even after adjustment for multiple covariates including comorbidities and other relevant clinical confounders.

Findings from our community-based retrospective cohort study indicate that HCMV antibody level was not a predictor of cardiovascular outcomes (cardiovascular death, ACS and MACCE) in individuals with no history of CVD at baseline. The lack of association with cardiovascular outcomes agrees with a previous study of 1612 individuals (aged 40–89 years) who participated in the 1981 Busselton Health Survey [20]. The authors found no associations between antibody levels and the development of coronary heart disease or stroke over a 17 year follow-up period [20]. A recent large study of 8531 individuals (aged 40–69 years) without prevalent CVD from the population-based UK Biobank data reported no associations between HCMV seropositivity or antibody levels with risk of incident CVD, ischemic heart disease or stroke during a mean follow-up period of 10.2 years [21]. The study was similar to ours but the HCMV seroprevalence rate of the cohort was lower (58% compared to 72%) and antibody levels were determined using one HCMV protein (pp28), whereas viral lysate was used in our assay, which allowed detection of antibodies reactive against all HCMV proteins. Furthermore, their study examined additional adjustments including physical activity, socioeconomic status and ethnicity in the multivariable analyses. The lack of association in our cohort reflects the low numbers of cardiovascular events during the 20-year follow-up period, and so our Cox regression models did not detect these small effects. A meta-analysis of studies involving individuals without pre-existing CVD at baseline may provide a sufficiently large sample size for insight and clarification of the relationship between HCMV infection and cardiovascular outcomes.

Although we did not investigate the mechanisms by which HCMV antibodies predict CVD progression and mortality, a number of direct and indirect effects have been proposed. Cells within the vascular wall including smooth muscle cells, endothelial cells and macrophages are primary cellular targets for HCMV replication [4], and viral antigens and nucleic acids have been detected in carotid atherosclerotic plaques [22,23]. However, it is now apparent that indirect effects of HCMV infection have significant health consequences. Mechanistically, sporadic reactivation of CMV throughout life can induce a chronic inflammatory state and accelerate immune ageing, resulting in diminished response to vaccination and increased susceptibility to infection, leading to increased risk of future cardiovascular events and mortality [24].

Discrepancies in the findings from different studies examining the association between HCMV infection and risk of CVD may be attributed to differences in the ages of the study cohorts, characteristics of participants at baseline (e.g., comorbidities), definition of main outcomes, various follow-up times and how the HCMV antibody was analysed (i.e., serostatus, antibody levels as a continuous variable or stratified into tertiles). Analysis of HCMV antibody levels rather than serostatus may be more informative as it reflects repeated reactivations and periods of viral replication that occur throughout life [10]. We have previously demonstrated higher HCMV antibody levels in renal transplant recipients with detectable CMV DNA in plasma and saliva [25]. Furthermore, antibody levels can be used to evaluate the relationship with vascular health. Levels of HCMV antibody were associated with flow-mediated dilation [26,27] and carotid intima-media thickness [28], both of which are early indicators of atherosclerosis and independent predictors of future cardiovascular events [29,30].

Our study has several limitations that have to be taken into account when interpreting the results. This includes the cross-sectional design, which can make determining causal relationships between HCMV antibody levels and 20-year risk of all-cause mortality more difficult. The contribution of changes in HCMV antibody levels over time (i.e., repeated measurements) warrants investigation in future studies. Furthermore, because the study population comprised of older Caucasian people, our findings may not be applicable to younger people or to other non-Caucasian populations due to differences in HCMV seroprevalence rates [5].

In summary, HCMV infection is associated with all-cause mortality over a 20-year period in a community-based cohort of adults without pre-existing CVD. Further larger studies are warranted to understand the mechanisms underpinning the complex relationship between HCMV infection and health outcomes in immunocompetent adults.

## Figures and Tables

**Table 1 viruses-14-02676-t001:** Baseline characteristics of Busselton Health Study participants (n = 2050).

Characteristic	Value
Age (yrs)	56.5 ± 11.2
Male	889 (43.4%)
Smoking status Never	1191 (58.1%)
Ex	639 (31.2%)
Current	220 (10.7%)
BMI (kg/m^2^)	26.6 ± 4.3
BP treatment	344 (16.8%)
Systolic BP (mmHg)	126.0 ± 17.0
Cholesterol (mmol/L)	5.79 ± 1.07
HDL cholesterol (mmol/L)	1.41 ± 0.40
Triglycerides (mmol/L)	1.37 ± 0.89
Glucose (mmol/L)	5.07 ± 1.29
CRP (mg/L)	3.26 ± 9.30
eGFR (mL/min)	68.1 ± 11.5
Diabetes	132 (6.4%)
COPD	190 (9.3%)
log HCMV IgG (AU/mL)	8.59 ± 4.12
log sCD14 (ng/mL)	14.2 ± 0.3
log sCD163 (ng/mL)	6.18 ± 0.38
All-cause death 5 years	59 (2.9%)
10 years	161 (7.9%)
20 years	448 (21.9%)
CVD death 5 years	22 (1.1%)
10 years	51 (2.5%)
20 years	152 (7.4%)
ACS 5 years	25 (1.2%)
10 years	63 (3.1%)
20 years	139 (6.8%)
MACCE 1 5 years	93 (4.5%)
10 years	236 (11.5%)
20 years	575 (28.0%)
MACCE 2 5 years	60 (2.9%)
10 years	138 (6.7%)
20 years	319 (15.6%)

Values are presented as mean ± standard deviation or n (%). ACS: acute coronary syndrome, BMI: body mass index, BP: blood pressure, COPD: chronic obstructive pulmonary disease, CRP: C-reactive protein, HDL: high density lipoprotein, CVD: cardiovascular disease, eGFR: estimated glomerular filtration rate, MACCE 1: major adverse cardiovascular and cerebrovascular events (composite of all-cause death, ACS, stroke and coronary artery revascularisation procedure), MACCE 2: major adverse cardiovascular and cerebrovascular events (composite of cardiovascular death, ACS, stroke and coronary artery revascularisation procedure).

**Table 2 viruses-14-02676-t002:** Hazard ratios (95% confidence intervals) for baseline HCMV antibody levels (log-transformed) for outcomes of all-cause death, CVD death, ACS, and MACCE at the 20-year follow-up.

	Model 1 ^3^		Model 2 ^4^		Model 3 ^5^		Model 4 ^6^	
	HR (95% CI)	*p*-value	HR (95% CI)	*p*-value	HR (95% CI)	*p*-value	HR (95% CI)	*p*-value
All-cause death	1.04 (1.01, 1.08)	0.013	1.04 (1.00, 1.07)	0.024	1.03 (1.00, 1.07)	0.039	1.04 (1.00, 1.07)	0.037
CVD death	1.03 (0.97, 1.09)	0.368	1.01 (0.96, 1.07)	0.668	1.01 (0.95, 1.07)	0.766	1.01 (0.95, 1.07)	0.757
ACS	0.99 (0.95, 1.03)	0.611	0.98 (0.94, 1.03)	0.452	0.98 (0.94, 1.03)	0.451	0.98 (0.94, 1.03)	0.451
MACCE 1 ^1^	1.03 (1.00, 1.05)	0.053	1.02 (1.00, 1.05)	0.104	1.02 (0.99, 1.05)	0.188	1.02 (0.99, 1.05)	0.127
MACCE 2 ^2^	1.01 (0.97, 1.04)	0.633	1.00 (0.97, 1.04)	0.797	1.00 (0.97, 1.04)	0.806	1.00 (0.97, 1.04)	0.811

^1^ composite of all-cause death, ACS, stroke and coronary artery revascularisation procedures; ^2^ composite of CVD death, ACS, stroke and coronary artery revascularisation procedures; ^3^ Model 1: Adjusted for sex and age; ^4^ Model 2: Model 1 plus additional adjustment for smoking, BMI, BP treatment, SBP, diabetes, cholesterol, HDL cholesterol, triglycerides, glucose and CRP; ^5^ Model 3: Model 2 plus additional adjustment for COPD and eGFR; ^6^ Model 4: Model 3 plus additional adjustment for sCD14 and sCD163. ACS: acute coronary syndrome, BMI: body mass index, BP: blood pressure, COPD: chronic obstructive pulmonary disease, CRP: C-reactive protein, CVD: cardiovascular disease, eGFR: estimated glomerular filtration rate, HDL: high density lipoprotein, SBP: systolic blood pressure.

## Data Availability

Restrictions apply to the availability of these data.

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
