# Peer review of "Long-Term Associations between Human Cytomegalovirus Antibody Levels with All-Cause Mortality and Cardiovascular Outcomes in an Australian Community-Based Cohort"

_viruses, 2022, doi:10.3390/v14122676_

Round 1

Reviewer 1 Report

In their study „Long-term associations between human cytomegalovirus antibody levels with all-cause mortality and cardiovascular outcomes in an Australian community-based cohort“, Silvia Lee and her colleagues analyze the association of HCMV antibody levels and the mortality in a retrospective cohort of 2050 study participants. The manuscript is in overall well written. However, I have several major points:

1) Line 131: “The prevalence of HCMV seropositivity in the cohort was 72%”. I may have missed it, but were HCMV seronegatives included or excluded in the study?

2) Is the seroprevalence itself (and not the CMV-specific IgG titers) associated with an increased mortality in the study cohort? It would be interesting to see, whether HCMV seronegatives have the lowest risk for an “all-cause death”, followed by individuals with low HCMV-specific antibody levels.

3) The authors need to further specify “all-cause death”, especially as the main differences were found in this cohort. It would be useful to list the most frequent causes of death in an additional supplementary table.

Minor points:

Line 38-39: The authors state in the introduction section that HCMV establishes an “asymptomatic latent infection in immunocompetent adults”. Symptomatic primary infections, reactivations and re-infections do, also if rarely, occur in immunocompetent individuals. I thus recommend to rephrase the sentence.

Table 1: The authors may consider to transfer the causes of death from the footnote directly into the table (e.g. MACCE 1 (all-cause death, ACS, stroke,…).

Line 98-99: As most readers of Virology may not be familiar with the role of sCD14 and sCD163, the authors should briefly explain why both markers were excluded in the study design.

Author Response

We thank the Reviewer for their critique of this work. Changes made are marked as track changes in the manuscript and are described below.

Reviewer 1

In their study „Long-term associations between human cytomegalovirus antibody levels with all-cause mortality and cardiovascular outcomes in an Australian community-based cohort“, Silvia Lee and her colleagues analyze the association of HCMV antibody levels and the mortality in a retrospective cohort of 2050 study participants. The manuscript is in overall well written. However, I have several major points:

1) Line 131: “The prevalence of HCMV seropositivity in the cohort was 72%”. I may have missed it, but were HCMV seronegatives included or excluded in the study?

SL: HCMV seronegatives were included in the analyses.

2) Is the seroprevalence itself (and not the CMV-specific IgG titers) associated with an increased mortality in the study cohort? It would be interesting to see, whether HCMV seronegatives have the lowest risk for an “all-cause death”, followed by individuals with low HCMV-specific antibody levels.

SL: We chose not to examine HCMV serostatus as most previous studies have used this stratification to examine association with all-cause death. Also, analysis of HCMV antibody levels as a continuous variable better reflects an individual’s cumulative viral burden over time.

3) The authors need to further specify “all-cause death”, especially as the main differences were found in this cohort. It would be useful to list the most frequent causes of death in an additional supplementary table.

SL: The top 10 leading causes of death have been included in a supplementary table.

Minor points:

Line 38-39: The authors state in the introduction section that HCMV establishes an “asymptomatic latent infection in immunocompetent adults”. Symptomatic primary infections, reactivations and re-infections do, also if rarely, occur in immunocompetent individuals. I thus recommend to rephrase the sentence.

SL: The sentence is reworded to acknowledge this.

Table 1: The authors may consider to transfer the causes of death from the footnote directly into the table (e.g. MACCE 1 (all-cause death, ACS, stroke,…).

SL: We feel that the Table is clearer to read without the details from the footnote.

Line 98-99: As most readers of Virology may not be familiar with the role of sCD14 and sCD163, the authors should briefly explain why both markers were excluded in the study design.

SL: I assume that the reviewer meant included and not excluded. sCD14 and sCD163 are soluble biomarkers of monocyte and macrophage activation. These markers were selected as these cells are the primary targets for HCMV and are a site of viral latency and persistence (Ref 4). We therefore addressed whether adjustment for these markers affected associations observed for HCMV. We have included this information (line 130-132).

Reviewer 2 Report

Major concerns

1.      - It seems that the calculations are all based on a single antibody measurement at the beginning of the study? The authors undoubtedly realize that these values can fluctuate of the life of an individual. Authors allude to this in discussion “The contribution of changes in HCMV antibody levels warrants investigation in future studies” but it is not clear to me that we are referring to the same thing.

2.      - The study seems potentially underpowered to detect cardiovascular association given that some of the associations of interest have previously required meta- analyses or much larger populations. This is alluded to in discussion but should be more explicitly acknowledged. Also the final conclusions about “not with ACS or MACCE (line 233-4)” needs to be softened, as the possibility of a missed association exists.

3.      - It is unclear to this reviewer why a commercially available ELISA was not used. A home grown ELISA was used instead, which in the end is fine, but details of these results are not presented. There is mention of a skewed distribution that should be shown both raw and log corrected. Performance of the ELISA should also be included, vis a vis false positive and negative, and sensitivity as well as the standard curve used to interpolate quantitation…also more elaboration on the relative quantitation. these data are central to the whole manuscript and must be provided.

4.      - It would be of interest to see whether there is an association of shed CD14 and CD163 with viral antibody….even if it didn’t influence the model these might have other effects that would be clinically interesting. Is there any correlation between estimated IgG and

Minor concerns

1.      Line 98 sCD14 and sCD163 are technically not cytokines – these are biomarkers.

Author Response

We thank the Reviewer for their critique of this work. Changes made are marked as track changes in the manuscript and are described below.

Reviewer 2

Major concerns

1.   - It seems that the calculations are all based on a single antibody measurement at the beginning of the study? The authors undoubtedly realize that these values can fluctuate of the life of an individual. Authors allude to this in discussion “The contribution of changes in HCMV antibody levels warrants investigation in future studies” but it is not clear to me that we are referring to the same thing.

SL: This is clarified (line 241).

2.   - The study seems potentially underpowered to detect cardiovascular association given that some of the associations of interest have previously required meta- analyses or much larger populations. This is alluded to in discussion but should be more explicitly acknowledged. Also the final conclusions about “not with ACS or MACCE (line 233-4)” needs to be softened, as the possibility of a missed association exists.

SL: We have amended the sentence to explicitly acknowledge that we do not have sufficiently large sample size to detect small effects. We have removed any mention of ACS or MACCE in the final conclusion, given the low number of events observed in our study population.

3.   - It is unclear to this reviewer why a commercially available ELISA was not used. A home grown ELISA was used instead, which in the end is fine, but details of these results are not presented. There is mention of a skewed distribution that should be shown both raw and log corrected. Performance of the ELISA should also be included, vis a vis false positive and negative, and sensitivity as well as the standard curve used to interpolate quantitation…also more elaboration on the relative quantitation. these data are central to the whole manuscript and must be provided.

SL: The in-house HCMV antibody ELISA has two advantages over commercially available ELISAs. It is able to detect all antibodies reactive against HCMV antigens in the lysate (this is not specified with most commercial kits). We are also able to run extensive serial dilutions to guarantee accurate determinations in the high range. We have included more details of the ELISA (line 97-103).

4.    - It would be of interest to see whether there is an association of shed CD14 and CD163 with viral antibody….even if it didn’t influence the model these might have other effects that would be clinically interesting. Is there any correlation between estimated IgG and

SL: The results (line 160-163) are modified to note the weak correlations between sCD14 and HCMV antibody levels (r=0.05, p=0.01) and for sCD163 and HCMV antibody levels (r=0.07, p=0.0004).

Minor concerns

  1. Line 98 sCD14 and sCD163 are technically not cytokines – these are biomarkers.

SL: This is corrected (line 104).

Round 2

Reviewer 1 Report

The authors have satisfactory answered most of my questions. However, I have two points left:

11)      The authors should add a short statement in the materials & methods or results section of their manuscript how HCMV seronegatives were included in the study design. Were they included with an HCMV antibody titer of zero?

22)      I am still not convinced, why the authors included the sCD14 and sCD163 biomarkers in their study design. Is the idea behind that high sCD14 and sCD163 serum/plasma levels indicate recent HCMV-reactivation events? If so, the authors should cite in line 128-129 at least one paper, which demonstrate an association between sCD14 and sCD163 levels and HCMV-reactivation events.

Author Response

We thank the reviewer for their comments. Our edits are noted below and are made in the text using track changes.

11)      The authors should add a short statement in the materials & methods or results section of their manuscript how HCMV seronegatives were included in the study design. Were they included with an HCMV antibody titer of zero?

SL: With our in-house ELISA, seronegative individuals have antibody levels between the range of 0-3475 AU/ml and these values were used for statistical analyses. The assay cut-off value (i.e 3475 AU/mL) was determined by the inclusion of nine known seronegative samples assessed by the ARCHITECT CMV IgG assay (Abbott Diagnostic Systems, Lake Forrest, IL) in our in-house ELISA. Samples achieving HCMV antibody levels greater than three standard deviations above the mean of the nine seronegative samples were defined as seropositive. These details are now included in the Materials and Methods (lines 99-102).

22)      I am still not convinced, why the authors included the sCD14 and sCD163 biomarkers in their study design. Is the idea behind that high sCD14 and sCD163 serum/plasma levels indicate recent HCMV-reactivation events? If so, the authors should cite in line 128-129 at least one paper, which demonstrate an association between sCD14 and sCD163 levels and HCMV-reactivation events.

SL: We have included sCD14 and sCD163 in the study design as HCMV reactivation throughout life results in an increase in antibody levels but also production of biomarkers reflecting monocyte immune activation. We have cited papers that found correlations between HCMV antibody levels or HCMV viremia (indicative of reactivation) with levels of sCD14 and sCD163 (lines 130-132).

References added (numbers 14-16)

  • Letendre S et al. Higher Anti-Cytomegalovirus Immunoglobulin G Concentrations Are Associated With Worse Neurocognitive Performance During Suppressive Antiretroviral Therapy. Clin Infect Dis 2018 67:770-777
  • Gomez-Mora E et al. Elevated humoral response to cytomegalovirus in HIV-infected individuals with poor CD4+ T-cell immune recovery. PLOS One 2017
  • Gianelle S et al. Presence of Asymptomatic CMV and EBV DNA in Blood of Persons with HIV Starting Antiretroviral Therapy are Associated with Non-AIDS Clinical Events. AIDS 2020 34:849-57.